# Transcriptome Sequencing and Metabolome Analysis Reveal Genes Involved in Pigmentation of Green-Colored Cotton Fibers

**DOI:** 10.3390/ijms20194838

**Published:** 2019-09-29

**Authors:** Shichao Sun, Xian-peng Xiong, Qianhao Zhu, Yan-jun Li, Jie Sun

**Affiliations:** 1The Key Laboratory of Oasis Eco-agriculture, Agriculture College, Shihezi University, Bei 5 Road, Shihezi 832003, China; shichaosun@aliyun.com (S.S.); xianpengxiongshzu@aliyun.com (X.-p.X.); 2CSIRO Agriculture and Food, GPO Box 1700, Canberra 2601, Australia; Qianhao.Zhu@csiro.au

**Keywords:** naturally colored cotton, pigment, transcriptome, metabolome, *Gossypium hirsutum* L.

## Abstract

Green-colored fiber (GCF) is the unique raw material for naturally colored cotton textile but we know little about the pigmentation process in GCF. Here we compared transcriptomes and metabolomes of 12, 18 and 24 days post-anthesis (DPA) fibers from a green fiber cotton accession and its white-colored fiber (WCF) near-isogenic line. We found a total of 2047 non-redundant metabolites in GCF and WCF that were enriched in 80 pathways, including those of biosynthesis of phenylpropanoid, cutin, suberin, and wax. Most metabolites, particularly sinapaldehyde, of the phenylpropanoid pathway had a higher level in GCF than in WCF, consistent with the significant up-regulation of the genes responsible for biosynthesis of those metabolites. Weighted gene co-expression network analysis (WGCNA) of genes differentially expressed between GCF and WCF was used to uncover gene-modules co-expressed or associated with the accumulation of green pigments. Of the 16 gene-modules co-expressed with fiber color or time points, the blue module associated with G24 (i.e., GCF at 24 DPA) was of particular importance because a large proportion of its genes were significantly up-regulated at 24 DPA when fiber color was visually distinguishable between GCF and WCF. A total of 56 hub genes, including the two homoeologous Gh4CL4 that could act in green pigment biosynthesis, were identified among the genes of the blue module that are mainly involved in lipid metabolism, phenylpropanoid biosynthesis, RNA transcription, signaling, and transport. Our results provide novel insights into the mechanisms underlying pigmentation of green fibers and clues for developing cottons with stable green colored fibers.

## 1. Introduction

Naturally colored cottons (NCCs) are varieties of cotton that produce fibers with non-white colors [1]. NCC accumulates its pigment during fiber development, requires no dyeing steps during fabric processing and manufacturing, there is thus no disposal of toxic dye waste [2,3,4], reducing manufacturing costs and being environmentally friendly [5,6]. However, the fiber color of NCCs are generally monotonous and unstable [7], which restricts large-scale production and utilization of NCCs.

Brown and green are the two most common colors observed in NCCs. Many studies have been carried out to understand the composition of pigments in colored fibers and the pathways involved in biosynthesis of pigments. The main pigment components in the brown-colored fiber (BCF) are proanthocyanidins or their derivatives generated from the flavonoid pathway. Several genes of the pathway that are related to pigment synthesis have been identified and characterized, such as genes encoding chalcone synthases (*CHS*), chalcone isomerase (*CHI*), flavanone 3-hydroxylases (*F3H*), flavonoid 3′, 5′-hydroxylases (*F3′5′H*), dihydroflavonol 4-reductases (*DFR*), leucoanthocyanidin reductases (*LAR*), anthocyanidin synthase (*ANS*), and anthocyanidin reductases (*ANR*) [4,8,9]. Genetically, at least 6 loci (*Lc1* to *Lc6*) have been reported to be associated with BCF [7]. A recent study identified the gene underlying the major BCF locus, *Lc1*, to be a R2R3-type MYB transcription factor encoding TRANSPARENT TESTA 2 (TT2) that regulates the expression of genes of the flavonoid pathway [10]. The pigment components of the green-colored fibers (GCF) are more complex than that of BCF. Sections of GCF have concentric osmiophilic layers which not found in BCF, osmiophilic layers that are each separated by cellulosic material. Chemical analysis of the isolated cell walls from green fibers confirmed the presence of suberin layers with 65% of their total monomers being 22-hydroxydocosanoic acid [11,12]. GCF are suberized and contain a large proportion of wax. The unidentified components of the wax could be separated into a colorless fluorescent fraction, and a yellow-pigmented fraction. The colorless fraction contains caffeic acid esterified to fatty acids (mainly ω-hydroxy fatty acids) and glycerol in a molar ratio of 4:5:5 [13,14,15]. Two caffeic-acid derivatives have been isolated from the yellow components of the GCF extract [1,16]. In plants, caffeic acid derivatives are produced by the phenylpropanoid biosynthesis pathway. When the enzymatic activity of phenylalanine ammonia lyase (PAL) is inhibited, in vitro cultured fibers from ovules of GCF variety remained white, and the colorless caffeic-acid derivatives and yellow components could no longer be detected [14]. Additionally, inhibition of fatty acid elongation activity results in discontinuous suberin layers and reduced lamellae [17]. Zhao and Wang [18] reported that the contents of flavonoids, including flavone and flavanols, were higher in GCF than in WCF.

In this study, we compared transcriptomes and metabolomes of GCF and WCF, explored the gene regulatory networks of pigmentation in GCF, and found differentially accumulated metabolites and differentially expressed genes related to biosynthesis of phenylpropanoids and flavonoids. Additionally, we used weighted gene co-expression network analysis (WGCNA) to identify a gene module highly related to the green fiber color at 24 DPA. The module contained two candidate hub genes (*Gh4CL4_At* and *Gh4CL4_Dt*) encoding 4-coumarate:Coenzyme A ligase (4CL). Our findings provided new insights into the molecular mechanisms responsible for pigment biosynthesis in GCF.

## 2. Results

### 2.1. Overview of the Metabolomic Profiling

To compare GCF and WCF metabolites, datasets obtained from quadrapole time of flight–mass spectrometry (QTOF–MS) by electrospray ionization (ESI^+^ and ESI^–^) were subjected to principal component analysis (PCA). The results showed that metabolites from different time points (12, 18 and 24 DPA) of GCF and WCF were clearly separated in the score plots, where the first principal component (PC1) was plotted against the second principal component (PC2). For the ESI+ mode, PC1 and PC2 represented 39.83% and 16.22% of the total variations, respectively (Figure 1A), and for the ESI^−^ mode, PC1 and PC2 accounted for 15.56% and 41.39% variations, respectively (Figure 1B). Plots from partial least squares discriminant analyses (PLS-DA) was further used to model the metabolite differences between GCF and WCF. At each time point, GCF and WCF were well separated in both ESI modes (Appendix A). These results suggested significant biochemical differences between GCF and WCF at 12–24 DPA.

To determine the effect of metabolites on fiber color, we did pairwise (GCF vs. WCF) comparison of the types of metabolites detected at each time point, and identified 852, 1011, and 1073 different metabolites in G12 vs W12, G18 vs W18 and G24 vs W24, respectively, representing a total of 2047 non-redundant metabolites. Of these metabolites, 95 were found to be commonly different at all three time points (Figure 1C). The number of different metabolites between GCF and WCF increased from 12 DPA to 24 DPA. At all three time points, more down-regulated metabolites and fewer up-regulated metabolites were observed in GCF than in WCF (Figure 1D).

### 2.2. Untargeted Metabolomic Analysis of Phenylpropanoid Metabolite Content

Significant metabolic difference was detected between GCF and WCF across all three time points. Kyoto Encyclopedia of Genes and Genomes (KEGG) analysis of the 2047 differential ions peaks showed that they were enriched for 80 pathways, including phenylpropanoid pathway, biosynthesis of plant secondary metabolites, and cutin, suberin, and wax biosynthesis pathway (Figure 2). In the phenylpropanoid pathway, 30 different metabolites (Appendix A) were detected between GCF and WCF, and 12 were up-regulated in G24 compared to W24. Sinapaldehyde was the most significantly up-regulated metabolite in G24, followed by caffeic acid, quercitrin, 5-hydroxyconiferaldehyde and ferulic acid. The amount of sinapaldehyde was >100 times higher in G24 than in W24 (Table 1).

Fourteen different metabolites involved in the biosynthesis of cutin, suberin, and wax were detected, and 10 were up-regulated in G24 compared to W24 (Table 2). These included three 22-carbon chain (C22) hydroxyalkanoic acids (22-hydroxydocosanoate, 22-oxo-docosanoate, docosanedioate), four 18-carbon chain (C18) hydroxyalkanoic acids (9,10-dihydroxystearate, 9,10-epoxy-18-hydroxystearate, cis-9,10-epoxystearic acid, 18-hydroxyoleate), and three 16-carbon chain (C16) hydroxyalkanoic acids (hexadecanedioate, 16-hydroxypalmitic acid, 16-oxo-palmitate). Docosanedioate was the most significantly up-regulated metabolite in G24, with an 11.85-fold increase compared to that in W24 (Table 2).

### 2.3. Global Transcriptome Changes During the Process of Fiber Pigmentation

To determine the global transcriptomic profile associated with fiber pigmentation, RNA-seq was performed using 12, 18, and 24 DPA RNA from green fiber cultivar C7 and its WCF near-isogenic line (C7-NIL). The Q30 percentage (sequences with sequencing error rate lower than 0.1%) was over 90%, and the average GC content of the RNA-seq reads was 44.7% (Appendix A). After filtering, approximately 50 million clean reads were retained for each sample. We found that 92%–94% of the clean reads could be mapped to the *G. hirsutum* reference genome [19], of which 83%–86% were uniquely mapped. Approximately 90% of the clean reads mapped to the 70,478 annotated genes. The mapped reads were used to calculate gene expression levels based on the fragments per kilobase of transcript per million mapped reads (FPKM) that were further used in analysis of differentially expressed genes (DEGs) [20].

Using the criteria of |log2 (fold change)| > 2 and Padj < 0.05, we identified a total of 8467 DEGs between GCF and WCF at the three time points, including 1443 at 12 DPA, 4278 at 18 DPA, and 5473 at 24 DPA. Between GCF and WCF, there were more up-regulated genes than down-regulated ones at 12 and 18 DPA, whereas there were slightly more down-regulated genes than up-regulated ones at 24 DPA (Figure 3A). A total of 543 genes (332 up-regulated and 198 down-regulated) were differentially expressed at all the three time points (Figure 3B). Regarding the number of DEGs between different time points of each genotype (i.e., GCF or WCF), the highest was observed in G18 vs. G12 while the least was observed in W18 vs W12 (Figure 3A,C). Overall, a total of 13,438 non-redundant DEGs were found in the 7 comparisons (Figure 3D, Appendix A).

### 2.4. Co-Expression Network Analysis Identified Pigmentation-Related Differentially Expressed Genes (DEGs)

To investigate the gene regulatory network during fiber development, and to identify specific gene modules that are associated with pigment formation, 13438 non-redundant DEGs were subjected to WGCNA. Modules were defined as clusters of highly interconnected genes, in which genes within the same cluster have high correlation coefficients. WGCNA analysis identified 16 distinct modules (labeled with different colors) shown in Figure 4A, in which major tree branches define the modules. The correlation coefficients between each module eigengene of the 16 distinct modules with each distinct sample (trait) are shown in Figure 4B (Appendix A). Notably, 5 module–trait relationships (blue-G24, purple-W18, green-W24, red-G12 and magenta-G18) were highly significant (*r* > 0.8, *p* < 10^−3^; Figure 4B).

Pigment accumulation in naturally colored fibers starts at about 20 DPA [21]. Identification of a G24-specific module (the blue module) was thus particularly interesting. The majority genes of the blue module were significantly up-regulated in G24 but weakly expressed in WCF (Figure 5A). Of these up-regulated genes, those that were also up-regulated in G18 are of particularly interest, as they might be associated with pigment formation and accumulation. We also constructed gene networks of the blue-module genes using WGCNA and identified 56 hub genes based on the criteria of K_ME_ (eigengene connectivity) ≥0.99 and edge weight value ≥0.5. These genes were found to be involved in lipid metabolism, phenylpropanoid biosynthesis, RNA transcription, signaling, and transport (Figure 5B, Table 3). Previous studies have reported that green cotton fiber pigments are mainly hydroxycinnamic acid and its derivatives, which is synthesised by the phenylpropanoid pathway in plants. In this pathway, *4CL* is the key gene which encodes 4-coumaric acid:coenzyme A ligase that catalyze hydroxycinnamic acids into corresponding CoA thiolesters and its derivatives [22]. Two of the candidate hub genes were homologs of *Gh4CL4* (*Gh_A10G0456* and *Gh_D10G0473*).

The 2705 DEGs of the blue module could be classified into 17 main groups/bins based on their annotated functions, including regulation of protein activity (10.22%), signaling (7.09%), transcriptional regulation (5.91%), transport (5.87%), cell wall (4.38%), and stress (4.31%) (Appendix A). KEGG analysis indicated that these DEGs were highly enriched in the following pathways: pentose and glucuronate interconversions (*p*-value = 5.46 × 10^−9^, 37 genes), fatty acid metabolism (*p*_value = 3.45 × 10^−7^, 31 genes), cutin, suberin, and wax biosynthesis (*p*_value = 6.32 × 10^−7^, 17 genes), fatty acid elongation (*p*-value = 3.75 × 10^−6^, 18 genes), phenylpropanoid biosynthesis (*p*_value = 1.07 × 10^−5^, 44 genes), pyruvate metabolism (*p*_value = 1.72 × 10^−5^, 29 genes), biosynthesis of unsaturated fatty acids (*p*_value = 1.53 × 10^−4^, 16 genes), fatty acid biosynthesis (*p*_value = 3.96 × 10^−4^, 16 genes), flavonoid biosynthesis (*p*_value = 5.72 × 10^−4^, 11 genes) and biosynthesis of secondary metabolites (*p*_value = 8.91 × 10^−4^, 167 genes) (Figure 5C). Both metabolomic and transcriptomic analysis revealed significant changes in the pathways related to biosynthesis of phenylpropanoids, cutin, suberin, and wax, we therefore further analyzed selected genes of these two pathways.

### 2.5. Phenylpropanoid Pathway

One of the focuses of the present study was to understand the mechanisms underlying green pigmentation in cotton fibers based on comparative transcriptome analysis. To this end, we first used real-time quantitative polymerase chain reaction (RT-qPCR) to validate the expression changes obtained based on the RNA-seq data. Twelve genes from the phenylpropanoid pathway were selected for validation, and a strong correlation between the RNA-seq and qRT-PCR data was observed, indicating the reliability of the RNA-seq data (Appendix A). We then checked the relationship between the expression change of genes and the accumulation of their corresponding metabolites, and found a co-occurred relationship in G24 for the major genes and their metabolites of the phenylpropanoid pathway. To understand the regulatory network of phenylalanine and flavonoid metabolic pathways between GCF and WCF, we carried out Pearson correlation tests between relative quantitative changes of metabolites and transcripts according to Gianoulis et al. [23]. The result showed that 26 transcripts had correlation (R^2^ > 0.4) with 12 metabolites (Appendix A).

The phenylpropanoid pathway is one of the most important pathways associated with plant secondary metabolism, which includes two important branches, the phenylalanine and flavonoid metabolic pathways [24]. As shown in Figure 6, many genes and metabolites of the phenylalanine and flavonoid biosynthesis pathways were up-regulated in G24, including *PAL* (5 DEGs), cinnamate-3-hydrolase (*C3H*, 1 DEG), *4CL* (4 DEGs), ferulate-5-hydroxylase (*F5H*, 2 DEGs), hydroxycinnamoyl-CoA shikimate/quinate hydroxycinnamoyl transferase (*HCT*, 3 DEGs), caffeoyl-CoA O-methyltransferase (*CCoAOMT*, 2 DEGs), *CHS* (3 DEGs), *F3H* (1 DEG), *F3′H* (1 DEG), *F3′5′H* (1 DEG), flavanol synthase (*FLS*, 1 DEG), and *LAR* (1 DEG).

Most of these genes were significantly up-regulated in both G18 and G24 (Table 4). The expression levels of four *4CL* genes, *Gh_A05G3997*, *Gh_A10G0456*, *Gh_D05G3934,* and *Gh_D10G0473*, were 8.11-, 4.02-, 3.45-, and 5.04-times higher in G24 than in W24, respectively, which explains the high accumulation of 3,4-dihydroxystyrene in G24 (Figure 6). Caffeic acid was produced from p-coumaric acid, catalyzed by C3H (*Gh_A13G2072*, a 2.10-fold upregulation in G24), consistent with the increased accumulations of caffeic acid in G24. Aldehyde dehydrogenase catalyzes conversion of coniferyl aldehyde into ferulic acid, and its encoding gene *ALDH* (*Gh_D07G0047*) was upregulated 3.74 times in G24, consistent with increased ferulic acid (4.52-fold) content in G24. HCT catalyzes conversion of coumaroyl-CoA into caffeoyl quinic acid. Three *HCT* genes were up-regulated in G24, consequently the content of caffeoyl quinic acid increased 3.99-fold. As a result, the end-product of the phenylpropanoid pathway, sinapaldehyde, increased by 104.36-fold in G24 (Figure 6). Significantly up-regulated expression of the genes encoding CHS (*Gh_A12G0367*, *Gh_D05G2280* and *Gh_D12G0299*), F3H (*Gh_D09G1969*), F3′H (*Gh_A10G0500*), and FLS that convers naringenin into quercitrin and leucocyanidin accounted for the significant differences of the metabolites catalyzed by these enzymes between GCF and WCF. *LAR* (*Gh_D12G1686*, 3.54-fold upregulation) catalyzes leucopelargonidin to afzelechin, and F3′5′H (*Gh_A05G0557*, 1.82-fold upregulation) is involved in the conversion of dihydrokaempferol into leucodelphinidin. Increased expression of these two genes consisted with high quantities of their catalyzed flavanols in G24 (Figure 6).

### 2.6. Lipid Metabolism Pathway

In the cutin, suberin, and wax biosynthesis pathway, all the detected metabolites, except hexadecanoic acid and (9Z)-octadecenoic acid, were increased in G24 compared to W24. Cytochrome P450 86A1 (CYP86A1) is a fatty acid ω-hydroxylase, and cytochrome P450 86B1 (CYP86B1) is required for biosynthesis of very long chain ω-hydroxyacid and α-, ω-dicarboxylic acid [25,26]. In G24, expression of two *CYP86A1* genes up-regulated 14.95- and 11.39-fold, and two *CYP86B1* genes up-regulated 7.36- and 6.05-fold. This could largely explain the high accumulation of 18-hydroxyoleate and 22-hydroxydocosanoate in G24. Omega-hydroxypalmitate O-feruloyl transferase (HHT) can catalyze the conversion of p-Coumaroyl-CoA and sinapoyl-CoA, but not feruloyl-CoA [27]. Notably, the four *HHT* genes were found to be significantly up-regulated in G24, consistent with significant increase of sinapaldehyde but no accumulation of 16-feruloyloxypalmitic acid in G24 compared to W24.

The GCF contains a large proportion of wax, and very long-chain fatty acids/alcohols provide precursors for wax synthesis [28]. Many genes involved in the formation of long-chain fatty acids, very long-chain fatty acids, alcohols, and wax were also up-regulated in GCF, such as β-ketoacyl CoA synthase (*KCS*), β-ketoacyl CoA reductase (*KCR*), fatty acyl-CoA reductase (*FAR*), and long chain acyl-CoA synthetase (*LACS*) (Appendix A).

## 3. Discussion

### 3.1. Identifying Pigmentation-Related Metabolites in Green-Colored Fiber (GCF)

Understanding the molecular mechanisms controlling pigment formation in GCF is of great importance for developing cotton varieties with stable green-colored fibers and high fiber quality as green fiber color is thought to be unstable and associated with low fiber quality. Cinnamic acid and its derivatives have been identified as GCF pigments [14,29]. Additionally, two caffeic acid derivatives have been isolated from green fibers, and were found to be positively correlated with the degree of green color in cotton fibers [1,16]. Here, we detected significant accumulation of two types of cinnamic acids (caffeic acid and ferulic acid), and five types of cinnamic acid derivatives (caffeoyl quinic acid, 3,4-dihydroxystyrene, coniferylaldehyde, 5-hydroxy coniferaldehyde, and sinapaldehyde) in GCF. In plants, cinnamic acid and its derivatives are produced through the phenylpropanoid pathway. The first step of the pathway involves PAL, which catalyzes the deamination of phenylalanine to generate cinnamic acid [30]. High PAL activity is associated with the accumulation of anthocyanins and other phenolic compounds in fruit tissues of several plant species [31,32,33]. In *Arabidopsis*, the double mutant *pal1/pal*2 produced yellow seeds, as the uncolored proanthocyanidins were unable to undergo polymerization and oxidation to produce tannin pigments [30,34]. Compared to WCF, GCF showed significantly up-regulated expression level of *PAL*. Treating the in vitro cultured ovules from green-colored cotton with 2-aminoindan-2 phosphonic acid, a PAL inhibitor, made them remaining white [14], suggesting that the green pigments are synthesized via the phenylpropanoid pathway. The fiber color of GCF and WCF was visually distinguishable at ~24 DPA (Figure 7), but significant expression changes of the phenylpropanoid biosynthesis genes could be observed at 18 DPA (Figure 6), indicating that 18–24 DPA could be the critical time window for color transition.

More than 1000 flavonoids have been identified to date, some of which play key roles as pigments, in pathogen resistance, and in protection against oxidative stress [35,36,37,38]. Previous study showed that pigments extracted from green fibers include flavone and flavonols [18]. We detected anthocyanin, flavone, flavanonol, and flavanols in GCF, supporting involvement of flavonoids in green cotton pigmentation. Consistent with this observation, many genes, including *CHS*, *F3H*, *F3′H*, *F3′5′H*, *FLS* and *LAR,* involved in flavonoid biosynthesis were significantly up-regulated in GCF (Figure 6). CHS catalyzes the first reaction in anthocyanin biosynthesis and is required for production of the intermediate chalcone, the primary precursor for all classes of flavonoids [39]. The *CHS* expression level was found to be closely correlated with the biosynthesis of flavonoid [40,41]. *F3′H* and *F3′5′H* also play critical roles in the flavonoid biosynthetic pathway. Overexpression of *F3′H* contributed to anthocyanin accumulation, and *F3′5′H* expression was essential for anthocyanin biosynthesis [42,43,44]. LAR is present in both the anthocyanin and flavanone biosynthetic pathways. LAR belongs to the reductase–epimerase–dehydrogenase family and the short-chain dehydrogenase/reductase superfamily. Each LAR has a specific carboxy-terminal domain which may have different substrate specificity [45]. Based on our metabolome and transcriptome data, LAR activity favors production of afzelechin and gallocatechin from leucopelargonidin and leucodelphinidin, rather than catechin synthesis (Figure 6). Thus, leucodelphinidin is the dominant anthocyanin in GCF. The dihydroflavonols represent a branch point in flavonoid biosynthesis, as they are the only intermediates in the production of both the colored anthocyanins, through dihydroflavonol (DFR), and the colorless flavonols, through flavonol synthase (FLS) [46]. As a result, there is competition for the dihydroflavonol substrate. Up-regulation of *FLS* would increase biosynthesis of flavanols that might lead to down-regulation of DFR and reduced accumulation of anthocyanin. This is consistent with the previous result showing that the *FLS* expression was inhibited in white-colored petunia, which resulted in accumulation of colored anthocyanin and consequently pink flowers [46]. We did not find significant differences in the amount of colored anthocyanidins between GCF and WCF, suggesting that GCF is not colored by anthocyanidins. Overall, our results suggest that the GCF pigments are composed of cinnamic acid and its derivatives, with contributions from colorless anthocyanins, flavonols, and flavanols.

### 3.2. Identifying Gene-Expression Modules Associated with Pigmentation in GCF

Based on WGNCA, we identified developmental-stage-specific gene expression modules associated with pigmentation in GCF (Figure 4A,B). The most intriguing one is the blue module associated with G24, in which the two homoeologous *Gh4CL4* genes [16] were identified as candidate hub genes (Figure 5B). 4CL is a key enzyme in the phenylpropanoid biosynthesis pathway, catalyzing the formation of CoA-esters of cinnamic acids and their derivatives, which are the substrates for biosynthesis of flavonoids, lignin, suberin, coumarin, and wall-bound phenolic compounds [22,47]. The result obtained in this study supported our previous finding that *Gh4CL4* might be involved in the metabolism of caffeic and ferulic residues to affect pigmentation in GCF, because *Gh4CL4* was preferably associated with these two substrates in GCF [16].

Notably, many genes involved in long-chain fatty acid, very long-chain fatty acid, alcohol, and wax synthesis were also enriched in G24 (Appendix A). Long-chain fatty acids, very long-chain fatty acids, and alcohols are precursors of wax synthesis. In *Arabidopsis*, KCS2 and KCS20 are required to add two carbons to C22 fatty acids during cuticular wax and root suberin biosynthesis [48]. Suppressing KCR activity results in a reduction of cuticular wax production [49]. Overexpression of *CER6* gene leads to increases epidermal wax in the stem [28]. Heterologous expression of the *FAR* gene can produce significant amounts of fatty alcohols in cuticle waxes of plant [50]. We found that all these genes were significantly up-regulated in GCF, consistent with their roles in accumulation of wax in GCF [15].

In G24-specific module (blue module), the hub genes (Table 3) of metal transporter related were also highly expressed in GCF. These include Fe^2+^ transport protein, copper-transporting ATPase, copper transporter, zinc transporter, metal transporter NRAMP, and heavy metal-associated isoprenylated plant protein (HIPP). The levels of Fe^2+^ and Cu^2+^ were higher in GCF than in WCF [51]. FeCl_2_ and CuSO_4_ have been used to treat green cotton fabric to deepen the shade, and Fe^2+^ seems to have a greater effect than Cu^2+^ [52]. It has been suggested that although non-ferrous metal ions, such as Cu^2+^ and Fe^2+^, are not components of pigments in GCF, they may chelate with pigment substances to form more stable structures that may alter the color of cotton fibers.

## 4. Materials and Methods

### 4.1. Plant Materials and Treatments

Upland cotton (*G. hirsutum*) cultivar Xincaimian No. 7 (C7) with GCF and C7-NIL were grown at the Shihezi University Experimental Station in Shihezi City (44°27′ N, 85°94′ E), Xinjiang Autonomous Region, China. GCF samples were collected from developing bolls at 12, 18, and 24 days post-anthesis (DPA), designated G12, G18, and G24, respectively. WCF samples were also collected at the same time and designated W12, W18, and W24 (Figure 7). The samples were frozen immediately in liquid nitrogen and stored at −80 °C until use.

### 4.2. Metabolite Extraction and Profiling

Fresh fiber samples (25 mg) were weighed, and put into Eppendorf tubes with 800 µL pre-cooled methanol and water solution in a 1:1 ratio, then grounded with a metal ball homogenizer (35 Hz for 5 min). The grounded samples were incubated at −20 °C for 2 h and centrifuged for 15 min at 4 °C at 2500 relative centrifugal field (RCF). A 550 µL aliquot of the supernatant was moved into a new Eppendorf tube for subsequent analysis.

A total of 30 samples (3 time points × 2 genotypes × 5 biological replicates) were prepared and analyzed according to the guidelines of the liquid chromatography–mass spectrometry (LC–MS) system. Firstly, all chromatographic separations were performed using an ultra-high performance liquid chromatography (UPLC) system (Waters, Milford, USA). An ACQUITY UPLC BEH C18 column (100 mm*2.1 mm, 1.7 μm, Waters, Milford, USA) was used for reversed phase separation. The column oven was maintained at 50 °C. The mobile phase contained solvent A (water + 0.1% formic acid), and solvent B (acetonitrile + 0.1% formic acid) with a flow rate of 0.4 mL/min. Gradient elution conditions were as follows: 0–2 min with 100% solvent A; 2–11 min with 0–100% solvent B; 11–13 min with 100% solvent B; 13–15 min, 0 to 100% solvent A. The injection volume for each sample was 10 µL.

A high-resolution tandem quadrapole time of flight (QTOF) mass spectrometer, Xevo G2 XS (Waters, UK), was used to detect metabolites eluted from the column. QTOF was operated in both positive electrospray ionization (ESI^+^) and negative electrospray ionization (ESI^−^) modes. For ESI^+^, the capillary and sampling cone voltages were set at 2 kV and 40 V, respectively. For ESI^−^, the capillary and sampling cone voltages were set at 1 kV and 40 V, respectively. Mass spectrometry data were acquired in the MSE centroid mode. TOF mass range was 50–1200 Da, and the scan time was 0.2 s. For the MS/MS detection, all precursors were fragmented using 20–40 eV, and the scan time was 0.2 s. During acquisition, the acquisition rate was set to 3 s to calibrate accuracy of mass measurements. Furthermore, in order to evaluate the stability of the LC-MS during acquisition, a quality control (QC) sample, which is a pool of all the samples, was acquired after every 10 samples.

### 4.3. Metabolite Identification and Quantification

Mass spectrum peaks were extracted using the commercial software Progenesis QI (version 2.2, Waters, Milford, USA). The metabolite structures, m/z values, retention time (RT), and the fragmentation pattern were identified based on the available metabolite databases. The mass ion peaks were selected using product ion scan during the first stage of mass spectrometry (MS1), and precursor ion scan during the second stage of mass spectrometry (MS2). The selected peaks were identified by MS2 fragment patterns. Total ion chromatograms (TICs) and extracted ion chromatograms (EICs or XICs) of QC samples that summarize the metabolic profiles of all samples were used to calculate the area under each peak.

Variable importance of the projection (VIP) scores of partial least-squares discriminant analysis (PLS-DA) was used to rank metabolites between the two upland cotton accessions. The VIP threshold was set to 1. In addition, ratio and t-test were also used as a univariate analysis to screen metabolites between samples. Samples with a ratio ≥ 1.2 or ≤ 0.83, *p*-value < 0.05 and VIP ≥ 1 were considered to be metabolites that were differentially present.

### 4.4. RNA Isolation and Illumina Sequencing

Total RNA was extracted from 12, 18 and 24 DPA fibers of C7 and C7-NIL using the Tiangen RNA Extraction Kits. Residue genomic DNA was degraded with DNase I (Promega, Beijing, China). RNA quantity and quality were determined by NanoDrop ND2000 spectrophotometer (NanoDrop Technologies, Wilmington, DE, USA), and Agilent Bioanalyzer 2100 system (Agilent Technologies, Palo Alto, CA, USA), respectively. RNA integrity was further determined by 1% agarose gel electrophoresis. A total of 3 µg total RNA per sample (in total 18 samples = 3 time points × 2 genotypes × 3 biological replicates) was used as input material for RNA-seq library preparation. RNA-seq (150 bp paired-end reads) was performed on an Illumina Hiseq 4000 platform. The whole set of annotated genes can be found in the National Center for Biotechnology Information (NCBI) SRA database (BioProject accession: PRJNA565616). Raw data (in FASTq format) were processed using Perl scripts to measure Q20, Q30, and GC content. After removing low-quality reads, the remaining cleaned data were used in all downstream analysis.

### 4.5. RNA Sequencing Data Analysis

The *G. hirsutum* reference genome and gene model annotation files were downloaded from https://www.cottongen.org (accessed on February 7, 2018) [19]. Indexes of the reference genome were built using Bowtie (version 2.2.3, Johns Hopkins University, Baltimore, Maryland, USA) and the paired-end clean RNA-seq reads were aligned to the reference genome using TopHat (version 2.0.12, Maryland University and California University, College Park and Oakland City, State of Maryland and California, USA) was used to count the number of reads mapped to each gene. The fragments per kilobase of transcript per million mapped reads (FPKM) of each gene was calculated and used to quantify the expression level of the annotated genes. Genes differentially expressed (DEG) between GCF and WCF (each with three biological replicates) were identified by DESeq packaged in R (1.18.0). DESeq can determine differential expression using a model based on negative binomial distribution. The resulting *p*-values were adjusted using the Benjamini and Hochberg’s approach to control the false discovery rate (FDR). Genes with an adjusted *p*-value < 0.05 were considered to be differentially expressed.

### 4.6. Assignment of MapMan Bins and Kyoto Encyclopedia of Genes and Genomes (KEGG) Pathways

The functions of DEGs were annotated based on their homology with *Arabidopsis* genes annotated by TAIR (The *Arabidopsis* Information Resource) and classified into MapMan bins using the Mercator pipeline for automated sequence annotation (http://mapman.gabipd.org/web/guest/app/mercator, accessed on March 5, 2018) [53]. Pathway enrichment was done using the KEGG orthology-based annotation system (KOBAS) based on the Kyoto Encyclopedia of Genes and Genomes (KEGG) database [54].

### 4.7. Gene Network Construction and Visualization

Co-expression networks were constructed using the WGCNA (v1.29, Department of Human Genetics, University of California, Los Angeles, CA, USA) package in R [55]. The modules were obtained using the automatic block-wise network construction approach. Modules were identified based on the default settings, with the exceptions of power, minModuleSize and merge CutHeight being set to 14, 30, and 0.25, respectively, and TOMType being selected. The eigengene value was calculated for each module and used to test for association with each sample. Total connectivity and intramodular connectivity (function soft Connectivity), K_ME_ (for modular membership, also known as eigengene-based connectivity), and *p*_value were calculated. Each of the module genes are shown in Appendix A. The networks were visualized using Cytoscape [56].

### 4.8. Real-Time Quantitative Polymerase Chain Reaction (RT-qPCR)

The same RNA samples used in RNA-seq were used in RT-qPCR. Three microliters of total RNA were used to synthesize cDNA by using oligo (dT) and M-MLV Reverse Transcriptase (Takara, Dalian, China) according to the manufacturer’s instructions. cDNAs were then loaded in a 96-well plate for qRT-PCR analysis with a Light Cycler^®^ 480 II system (Roche, Switzerland) using the Power SYBR Green PCR Master Mixture (Roche, Switzerland), 10 µl reactions contained 1 µL of cDNA, 100 nM of each pair of target primers and 5 µL of SYBR Green PCR Master Mix. PCR conditions were as follows: 95 °C for 5 min, 40 cycles of 94 °C for 10 s, 60 °C for 10 s, and 72 °C for 10 s. Relative gene expression levels were analyzed according to the 2^ΔΔ*C*t^ method. The internal normalization gene was *GhHistone 3*. Primers were designed using National Center for Biotechnology Information PrimerBLAST tools (Available online: http://www.ncbi.nlm.nih.gov/tools/primer-blast/, accessed on March 27, 2018). Product specificity and reaction efficiencies were verified for each primer pair. Primer pairs are listed in Appendix A.

## Figures and Tables

**Figure 1 ijms-20-04838-f001:**
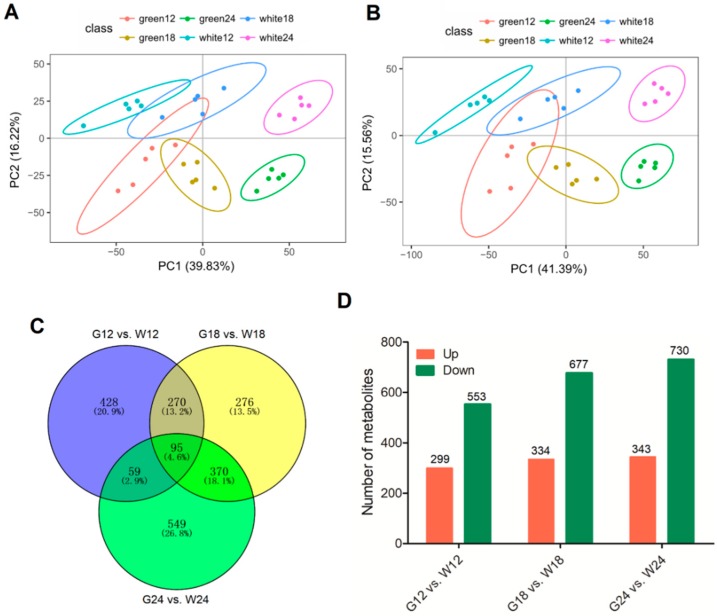
Comparison of metabolites from different developmental stages of cotton fibers. (**A**) Principal component analysis (PCA) score plot derived from metabolite ions acquired using the electrospray ionization positive ion mode (ESI^+^). (**B**) PCA score plot derived from metabolite ions acquired using the electrospray ionization nagative ion mode (ESI^−^). (**C**) Venn diagram showing different metabolites identified between green-colored fiber (GCF) and white-colored fiber (WCF). (**D**) The number of up- (red) and down-regulated (green) metabolites in each comparison.

**Figure 2 ijms-20-04838-f002:**
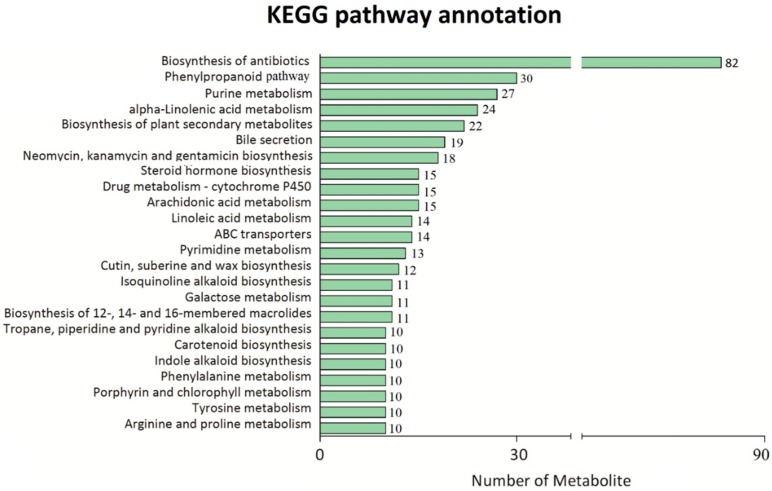
Kyoto Encyclopedia of Genes and Genomes (KEGG) pathway annotation of all the different metabolites.

**Figure 3 ijms-20-04838-f003:**
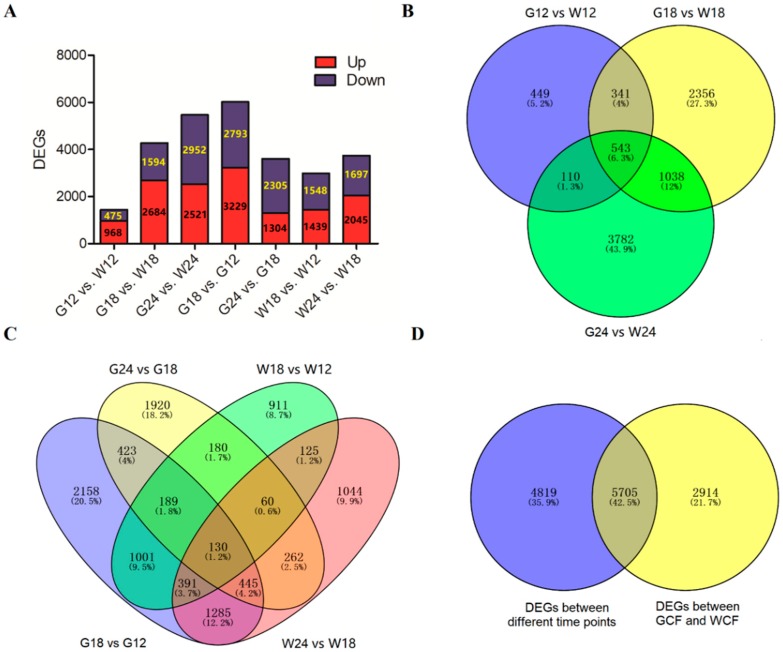
Differentially expressed genes (DEGs) between GCF and WCF. (**A**) The total number of DEGs identified in each comparison. (**B**) Venn diagram showing DEGs between GCF and WCF in the three time points. (**C**) Venn diagram showing DEGs between different time points in GCF or WCF. (**D**) The total number of non-redundant DEGs between different time points as well as between GCF and WCF.

**Figure 4 ijms-20-04838-f004:**
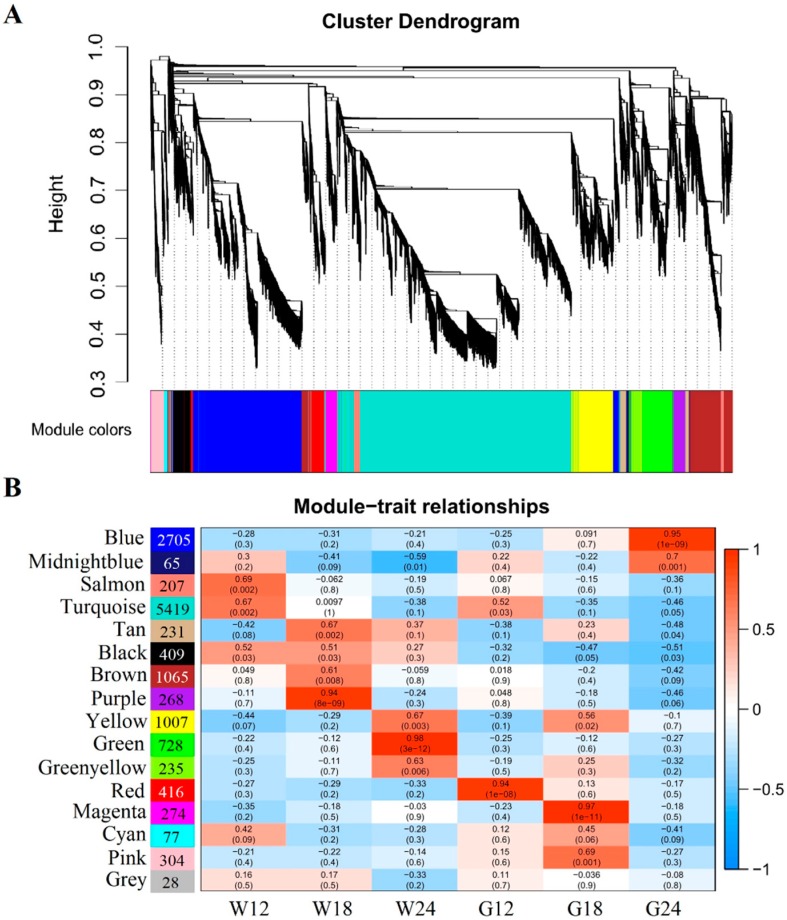
Weighted gene co-expression network analysis (WGCNA) of DEGs between GCF and WCF. (**A**) Hierarchical cluster tree showing co-expression modules identified by WGCNA. Each leaf in the tree represents one gene. Each major tree branch represents a distinct module, in total, there were 16 modules labeled by different colors. (**B**) Module-sample association relationships. Each row corresponds to a module, labeled by the same color as in (A). The number of genes in each module is shown next to the module name. Each column corresponds to a specific tissue. The correlation coefficient and *p*-value between the module and the sample or tissue are shown at the row-column intersection.

**Figure 5 ijms-20-04838-f005:**
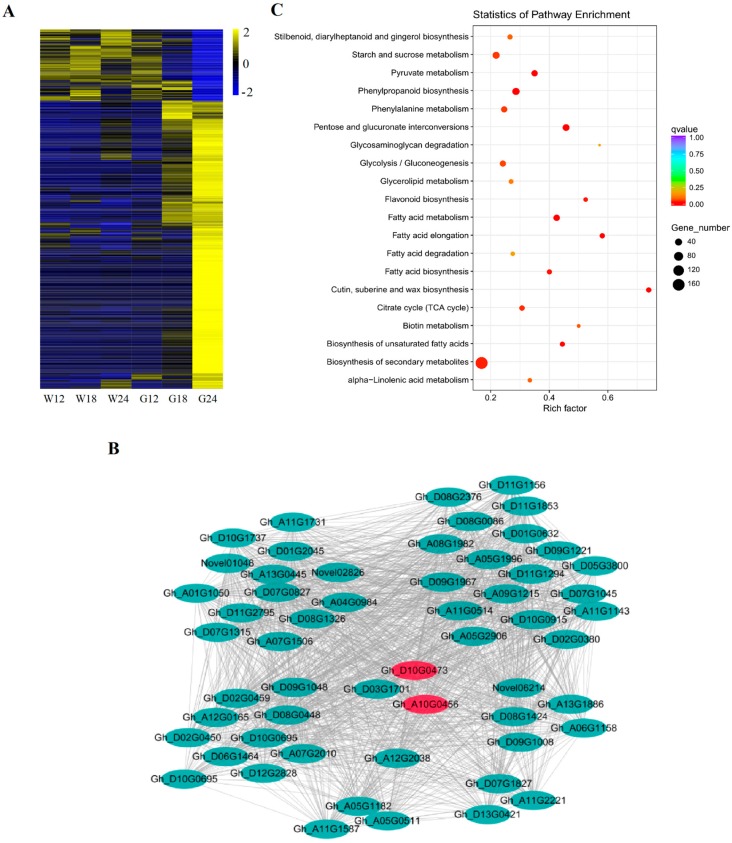
Co-expression network analysis of a stage-specific module. (**A**) Heatmap showing genes in the blue module that were over-expressed at G24. (**B**) Correlation networks of hub genes in the blue module. The two homoeologous *Gh4CL4* genes are shown in red. (**C**) The enriched KEGG pathways of the blue module genes.

**Figure 6 ijms-20-04838-f006:**
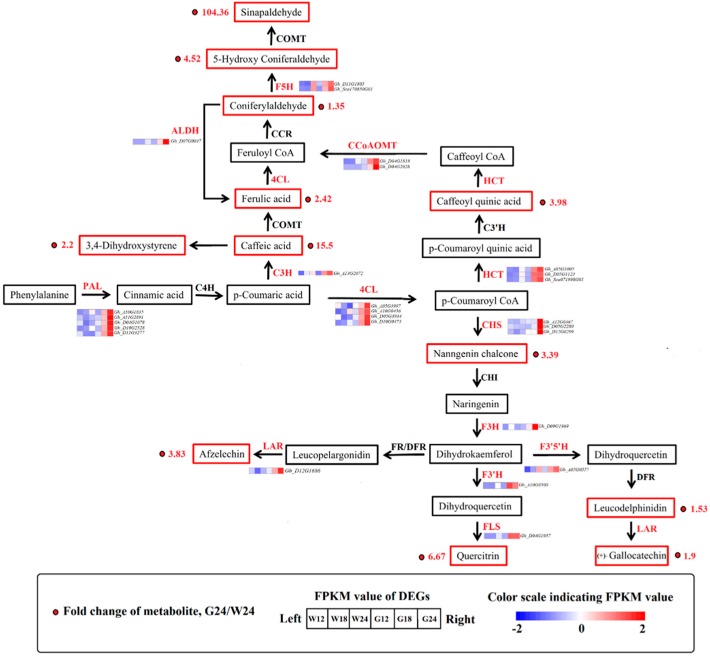
Transcript and metabolic profiling of genes in the phenylpropanoid and flavonoid biosynthetic pathways in cotton. *PAL*, phenylalanine ammonia-lyase; *4CL*, 4-coumarate CoA ligase; *C3H*, cinnamate 3-hydroxylase; *ALDH*, Aldehyde dehydrogenase; *HCT*, shikimate O-hydroxycinnamoyltransferase; *CCoAOMT*, caffeoyl-coenzyme A O-methyltransferase; *CHS*, chalcone synthase; *F3H*, flavanone 3-hydroxylase; *F3′H*, flavanoid 3′-hydroxylase; *DFR*, dihydroflavonol 4-reductase; *FLS*, flavonol synthesis; *LAR*, leucocyanidin reductase.

**Figure 7 ijms-20-04838-f007:**
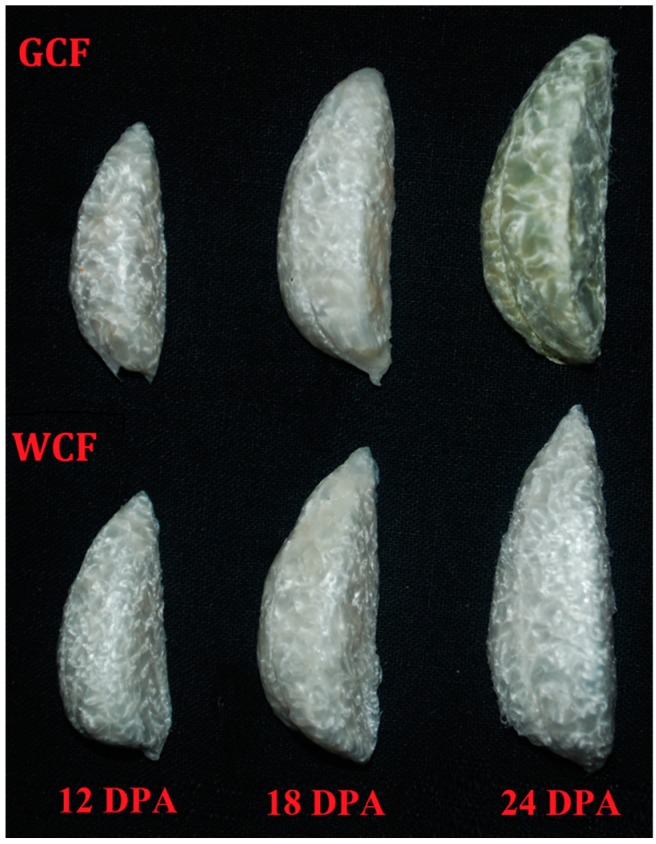
Fiber phenotypes of the green-colored fibers (GCF) and white-colored fibers (WCF) at 12, 18, and 24 days post-anthesis (DPA).

**Table 1 ijms-20-04838-t001:** Upregulated phenylpropanoids in G24 compared to W24.

Component	Metabolite Name	Log_2_ (Fold Change)	VIP	Exact Mass (m/z)
		(G24/W24)		
Intermediates	Caffeic acid	16.06	3.55	179.0336
	Ferulic acid	4.52	2.86	193.049
	Sinapaldehyde	104.36	3.46	207.0648
	3,4-Dihydroxystyrene	2.21	2.1	136.0529
	Coniferyl aldehyde	1.35	1.12	161.0601
	5-Hydroxyconiferaldehyde	4.52	2.86	177.055
	Caffeoyl quinic acid	3.98	2.2	355.0999
Flavone	Naringenin chalcone	3.39	1.44	271.0596
Flavanol	(+)-Gallocatechin	1.9	1.2	305.066
Flavonol	Afzelin	3.83	2.24	455.0991
	Quercitrin	6.67	2.64	449.1087
Anthocyanin	Leucodelphinidin	1.54	1	323.0751

VIP: Variable Importance in the Projection.

**Table 2 ijms-20-04838-t002:** Upregulated metabolites of the cutin, suberin, and wax biosynthesis pathway in G24 compared to W24.

Metabolite Name	Log_2_ (Fold Change)	VIP	Exact Mass (m/z)
	(G24/W24)		
Docosanedioate	11.85	2.5	369.2993
22-Oxo-docosanoate	1.85	1.37	353.3067
22-Hydroxydocosanoate	8.82	2.38	355.3204
9,10-Dihydroxystearate	3.03	2	315.2526
9,10-Epoxy-18-hydroxystearate	1.65	1.34	313.2365
cis-9,10-Epoxystearic acid	1.72	1.16	297.2418
18-Hydroxyoleate	2.01	1.61	298.2513
Hexadecanedioate	2.56	1.95	285.2057
16-Hydroxypalmitic acid	1.54	1.2	271.2261
16-Oxo-palmitate	1.45	1.01	269.2104

**Table 3 ijms-20-04838-t003:** List of the blue module hub genes.

**Gene ID**	**Description**	**K_ME_ Value**
**Lipid Metabolism**		
Gh_D11G1853	Epoxide hydrolase 3, EPHX3	0.997
Gh_A11G1143	Non-specific lipid-transfer protein-like protein, At2g13820	0.997
Gh_D02G0380	Lipid binding protein	0.997
Gh_D09G1221	Fatty acyl-CoA reductase, FAR	0.997
Gh_A05G1996	Probable glucan endo-1,3-beta-glucosidase 1, BG1	0.997
Gh_D08G0086	Fatty acyl-CoA reductase, FAR	0.995
Gh_D11G1294	Non-specific lipid-transfer protein-like protein, At2g13820	0.995
Gh_A08G1982	Sterol 3-beta-Glucosyltransferase, UGT80A2	0.995
Gh_A11G0514	Diacylglycerol O-acyltransferase 2, DGAT2	0.994
Gh_D01G0632	1-acyl-sn-glycerol-3-phosphate acyltransferase, AGPAT	0.993
Gh_D01G2045	Probable glycosyltransferase, At5g03795	0.993
Gh_D05G3800	Probable glucan endo-1,3-beta-glucosidase, BG1	0.992
Gh_A13G0445	Lipid binding protein	0.992
Gh_A09G1215	Fatty acyl-CoA reductase, FAR	0.992
Gh_D10G0915	GDSL glycine (G), aspartic acid (D), serine (S) and leucine (L) esterase/lipase, At2g23540	0.991
Gh_D09G1967	Xyloglucan Galactosyltransferase KATAMARI1 homolog, Os03g0144800	0.991
Gh_D08G2376	Sterol 3-beta-Glucosyltransferase, UGT80A2	0.991
Gh_D11G1156	Triacylglycerol lipase	0.991
Gh_D07G1045	GDSL esterase/lipase, At5g22810	0.990
Gh_A05G2906	Lipid transfer-like protein, VAS	0.990
**Phenylpropanoid Biosynthesis**		
Gh_A10G0456	4-coumarate-CoA ligase, 4CL	0.996
Gh_D10G0473	4-coumarate-CoA ligase, 4CL	0.995
Gh_D03G1701	Caffeoylshikimate esterase, CSE	0.991
**RNA**		
Gh_A13G1886	Scarecrow-like protein, SCL3	0.995
Novel06214	NAC domain protein	0.994
Gh_D08G1424	Probable WRKY transcription factor 43, WRKY43	0.994
Gh_D09G1008	LOB domain-containing protein 1, LBD1	0.993
Gh_A06G1158	Putative Myb family transcription factor, At1g14600	0.991
**Gene ID**	**Description**	**K_ME_ Value**
**Protein**		
Gh_A05G0511	Aspartyl protease family protein 2, NEP2	0.998
Gh_A05G1182	RING-H2 finger protein, ATL3	0.991
Gh_A11G1587	Aspartic proteinase-like protein, At5g10080	0.990
**Signalling**		
Gh_A11G2221	Cysteine-rich repeat secretory protein, CRRSP3	0.992
Gh_D13G0421	LRR receptor-like serine/threonine-protein kinase, LRR–RLK	0.992
Gh_D07G1827	Receptor-like protein kinase, FER	0.991
**Transport**		
Gh_A12G0165	Nucleobase-ascorbate transporter, NAT	0.997
Gh_D08G0448	Heavy metal-associated isoprenylated plant protein, HIPP	0.996
Gh_A07G2010	Aquaporin, SIP	0.994
Gh_D02G0459	Protein NRT1/ PTR family, NPF	0.992
Gh_D02G0450	Phosphate transporter, PHO	0.992
Gh_D06G1464	Aquaporin, SIP	0.992
Gh_D10G0695	ADP,ATP carrier protein 1, chloroplastic, AATP	0.991
Gh_D12G2828	Nucleobase-ascorbate transporter, NAT	0.991
Gh_D10G0695	ADP,ATP carrier protein 1, chloroplastic, AATP	0.991
Gh_D09G1048	ABC transporter G family member 23, ABCG23	0.990
**Hormone**		
Gh_A12G2038	Probable aminotransferase 10, ACS10	0.990
**Others**		
Gh_A07G1506	Pentatricopeptide repeat-containing protein, At3g22150	0.994
Gh_D10G1737	Uncharacterized protein	0.994
Gh_D08G1326	Uncharacterized protein	0.994
Gh_A04G0984	Periaxin, Prx	0.993
Novel01048	Uncharacterized protein	0.992
Gh_A11G1731	Uncharacterized protein	0.992
Gh_D07G0827	Condensin complex subunit	0.992
Gh_D07G1315	Uncharacterized protein	0.992
Novel02826	Uncharacterized protein	0.991
Gh_A01G1050	Uncharacterized protein	0.991
Gh_D11G2795	Uncharacterized protein	0.990

**Table 4 ijms-20-04838-t004:** List of the phenylpropanoid pathway genes.

	Log2 (Fold Change G/W)			Annotation	Symbol
	12 DPA	18 DPA	24 DPA		
Gh_A10G1835	0.74938	3.1288	3.1633	Phenylalanine ammonia lyase	*PAL*
Gh_A11G2891	0.27697	5.7791	6.3045		*PAL*
Gh_D04G1078	0.18112	4.0366	5.2906		*PAL*
Gh_D10G2528	0.76809	3.1311	3.7176		*PAL*
Gh_D11G3277	−0.0074771	6.0276	9.165		*PAL*
Gh_A05G3997	−0.49978	3.4355	8.1142	4-coumarate:CoA ligase	*4CL*
Gh_A10G0456	3.8493	6.0506	4.0219		*4CL*
Gh_D05G3934	−0.23208	2.4448	3.4508		*4CL*
Gh_D10G0473	1.5417	4.7604	5.0489		*4CL*
Gh_A13G2072	0.14536	1.6208	2.1007	Coumarate 3-hydroxylase	*C3H*
Gh_A05G1005	0.5165	3.8158	3.4893	Shikimate/quinate hydroxycinnamoyl transferase	*HCT*
Gh_D05G1123	0.55687	3.5151	2.4623		*HCT*
Gh_Sca071998G01	0.7392	4.052	3.1351		*HCT*
Gh_D04G1818	4.3784	5.9612	3.5403	Caffeoyl CoA O-methyltransfersae	*CCoAOMT*
Gh_D04G2028	Inf	Inf	Inf		*CCoAOMT*
Gh_D07G0047	0.90754	4.2483	3.475	Aldehyde dehydrogenase	*ALDH*
Gh_D11G1805	2.5784	7.8403	1.8926	Ferulate 5-hydroxylase	*F5H*
Gh_Sca170850G01	1.1641	Inf	1.5256		*F5H*
Gh_A12G0367	1.5724	2.819	4.867	Chalcone and stilbene synthase family protein	*CHS*
Gh_D05G2280	−1.449	1.9241	5.4836		*CHS*
Gh_D12G0299	1.2695	3.4081	4.445		*CHS*
Gh_D09G1969	0.69249	0.86727	4.5918	Flavanone 3-hydroxylase	*F3H*
Gh_A10G0500	0.40181	4.5526	1.7541	Flavonoid 3′-hydroxylase	*F3′H*
Gh_A05G0557	2.5078	3.5865	1.8235	Flavonoid 3′,5′-hydroxylase	*F3′5′H*
Gh_D04G1975	0.26368	−3.009	0.10962	Flavonol synthase	*FLS*
Gh_D12G1686	0.30587	2.7314	3.453	Leucoanthocyantin reducase	*LAR*

Inf represents that the gene is expressed in GCF, but no expression in WCF.

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
