# Peer review of "Transcriptome Sequencing and Metabolome Analysis Reveal Genes Involved in Pigmentation of Green-Colored Cotton Fibers"

_ijms, 2019, doi:10.3390/ijms20194838_

Round 1

Reviewer 1 Report

Sun et al describe the longitudinal transcriptomic and metabolomic analysis of white and green cotton fibers. They report changes in metabolites consistent with differences in color, as well as and genes that may be responsible. While all of the results are correlative, this study represents a significant amount of work and data that add value to the field. The major shortcoming is one of under-analysis – the analyses don’t take full advantage of the time points or paired metabolomics and transcriptomics data to model changes over time or link transcripts to metabolites in a principled way. Howevever, I don't see this as sufficient reason to delay publication.

The major change needed is to cite where the data will be publicly available upon publication.

Major points:

The analyses don’t utilize the three time points, rather, every description is either pair-wise comparison or aggregating all differences. This seems like a missed opportunity.

Moreover, the metabolites and genes are treated like independent datasets. There are known methods to relate the changes in genes to metabolites, e.g. compound context analysis by Gianoulis et al. https://journals.plos.org/plosgenetics/article?id=10.1371/journal.pgen.1002558 .

Minor points:

48: authors state that “GCF are more complex than BCF”, but the lines that follow do not make comparisons to BCF.

Reviewer 2 Report

Congrats for this piece of work. I only detected a few issues deserving attention that I explain below:

1.Major issues:

- FIGURE 6 IS MISSING

At the beginning I thought that it was my mistake in figure S2 but then I realized that the metabolic profiling is also missing. I couldn't evaluate this part of the results.

- There is no reference of the availability of the data. Are they deposited in a public database?

2.Minor issues:

- line 167: After 0.5 I would use ". These genes..." instead of ", these genes...".

- line172-173. "Therefore, two of the candidate hub genes were homoeologs of Gh4CL4(..."

First: "homologs" is misspelled.

Second: Where they?In terms of sequence homology?How do you infer that from WGCNA? If it is not a conclusion but it is a fact that they are orthologs/homologs, just remove "Therefore".

- Line 204: Figure 6 instead of Figure6.

- Line 248: The GCF are contain...(I guess you meant contain, but choose ).

- Line 253: You say you couldn't detect the metabolites corresponding to the up-regulated pathway due to inappropriate methods. I missed some discussion about that (why QTOF is not appropriate and which method could be used to validate the transcriptome data at metabolite level).

Round 2

Reviewer 2 Report

My questions to the authors were finally answered.